# Analysis of the Association between Running Performance and Game Performance Indicators in Professional Soccer Players

**DOI:** 10.3390/ijerph16204032

**Published:** 2019-10-21

**Authors:** Toni Modric, Sime Versic, Damir Sekulic, Silvester Liposek

**Affiliations:** 1Faculty of Kinesiology, University of Split, 21000 Split, Croatia; toni.modric@yahoo.com (T.M.); simeversic@gmail.com (S.V.); 2HNK Hajduk Split, 21000 Split, Croatia; 3University of Maribor, 2000 Maribor, Slovenia

**Keywords:** GPS, football, accelerations, decelerations, efficacy

## Abstract

Running performance (RP) and game performance indicators (GPI) are important determinants of success in soccer (football), but there is an evident lack of knowledge about the possible associations between RP and GPI. This study aimed to identify associations between RP and GPI in professional soccer players and to compare RP and GPI among soccer playing positions. One hundred one match performances were observed over the course of half of a season at the highest level of national competition in Croatia. Players (mean ± SD, age: 23.85 ± 2.88 years; body height: 183.05 ± 8.88 cm; body mass: 78.69 ± 7.17 kg) were classified into five playing positions (central defenders (n = 26), full-backs (n = 24), central midfielders (n = 33), wide midfielders (n = 10), and forwards (n = 8). RP, as measured by global positioning system, included the total distance covered, distance covered in five speed categories (walking, jogging, running, high-speed running, and maximal sprinting), total number of accelerations, number of high-intensity accelerations, total number of decelerations, and number of high-intensity decelerations. The GPI were collected by the position-specific performance statistics index (InStat index). The average total distance was 10,298.4 ± 928.7 m, with central defenders having the shortest and central midfielders having the greatest covered distances. The running (r = 0.419, *p* = 0.03) and high-intensity accelerations (r = 0.493, *p* = 0.01) were correlated with the InStat index for central defenders. The number of decelerations of full-backs (r = −0.43, *p* = 0.04) and the distance covered during sprinting of forwards (r = 0.80, *p* = 0.02) were associated with their GPI obtained by InStat index. The specific correlations between RP and GPI should be considered during the conditioning process in soccer. The soccer training should follow the specific requirements of the playing positions established herein, which will allow players to meet the game demands and to perform successfully.

## 1. Introduction

Soccer is a highly complex team sport with changing dynamics and multistructural movements played by two teams. Each team consists of 10 outfield players and a goalkeeper and the final game achievement depends directly on the performance of all 11 players [1,2]. Therefore, performance analysis is crucial in the evaluation of players’ achievement [3]. The global popularity of soccer has led to the implementation of scientific and technological knowledge in everyday use, and this is particularly evident within the field of performance analysis. One of the important aspects of performance analysis is termed “running performance”, which is nowadays mostly evidenced by global positioning software systems (GPS) [4]. 

GPS technology is known to be highly applicable in evaluation of mobility and physical activity patterns within the field of public health [5,6,7]. With the improvement of their accuracy/precision, design, usability and safeness (Figure 1), the GPS-based devices are becoming prevalent even in competitive sports, including soccer [8,9]. 

Specifically, GPS allows collecting data about players’ running performance, such as the total distance covered, the distance covered at different intensities (i.e., speeds), and the number of accelerations and decelerations. Studies conducted so far have provided playing-position-specific evidence with regard to running at different intensities, with midfielders covering the largest total distance and wingers performing the most high-intensive sprints [10]. Furthermore, match running performance in Brazilian professional soccer players indicated that winning teams, home playing teams and teams that play against “weaker” opponents had the greatest total distance covered [11]. A study performed with <21- and <18-year old soccer players found that a 3–5–2 formation elicited the highest total distance, with a 4–2–3–1 formation eliciting the highest number of accelerations and decelerations [12]. The results of the previously cited studies that used GPS technology as a measurement tool were generally consistent with those from investigations where authors used different video-based computerized match analysis systems in the evaluation of players’ running performances [13,14,15,16,17].

Game performance indicators are another set of variables that are used in performance analysis in soccer. Basically, game performance indicators are defined as a “selection and combination of variables that define some aspect of performance and that help achieve athletic success” [18]. The most frequently used game performance indicators are passes, shots, crosses, dribbles, challenges etc. [19]. Currently, numerous video-based platforms that track performance indicators of soccer players are available (InStat, Optasport, Wyscout). Such platforms quickly and accurately provide a large range of data about game performance indicators, allowing the simultaneous analysis of the physical efforts, movement patterns, and technical actions of players, both with and without the ball [20,21,22].

Previous studies conducted in the field of performance analysis in soccer found that both the physical (i.e., total distance covered, high-intensity running, accelerations and decelerations) and technical–tactical performances (i.e., shots, crosses, challenges, and dribbles) of players were correlated with specific conditions such as match outcome (win/draw/loss), match location (home/away), type of match (league/cup/friendly), and strength of the opponent team [19,23,24,25,26,27,28]. Situational variables, such as ball possession, total shots, shots on target, crosses, dribbles, clearances, challenges, and interceptions, and their influence on technical–tactical parameters were mostly evaluated by the variation of counts of technical match actions, which include shots, passing, tackles, aerial duels, and dribbles [18,27,29,30]. Briefly, situational variables that discriminate among winning, drawing and losing were mostly those related to ball possession and offensive actions (e.g., total shots, shots on goal, and crosses) [29,30], while some studies found that indicators of defensive efficacy (e.g., interceptions, clearance, and aerial challenges) were the variables most related to the match outcome [27].

Although game performance indicators and running performance are often investigated separately, to the best of our knowledge, there is no study that simultaneously observed both groups of performance variables during official soccer matches. Additionally, there is no information about the relationship that may exist between these two groups of variables. Therefore, the aim of this study was to identify possible associations that may exist between running performance and game performance in professional soccer players. Additionally, running performance and standard soccer performance variables were compared among playing positions. Authors were of the opinion that a study of this type would allow a better understanding of the relationships that exist between running performance and game performance indicators and that such understanding would therefore improve the applicability of both sets of variables in soccer training and competition.

## 2. Materials and Methods 

### 2.1. Participants and Design

The participants in this study were professional soccer players from Croatia (mean ± SD, age: 23.85 ± 2.88 years; body height: 183.05 ± 8.88 cm; body mass: 78.69 ± 7.17 kg), and all were members of one team competing at the highest national. Players were observed over one competitive half season, resulting in 101 match performances which were used as cases for this study. All data were collected during 14 matches of the Croatian Soccer League 2018/2019 season, and for the purpose of this study only the results of those players who participated in the whole game were analyzed. Players were classified in five groups based on playing positions: central defenders (CD; n = 26), full-backs (FB; n = 24), central midfielders (CM; n = 33), wide midfielders (WM; n = 10), and forwards (FW; n = 8), as suggested previously [10]. Sociodemographic and anthropometric data of observed players are presented in Table 1. In the observed half-season, the team played seven home and seven guest matches, with three wins, eight draws and three losses. At the end of the observed half-season, the team ranked 6th of 10 teams which competed in Croatian Soccer League. The investigation was approved by Ethical Board of the University of Split, Faculty of Kinesiology, Split, Croatia (approval number: 2181-205-02-05-19-0020).

### 2.2. Procedures

The variables in this study were two sets of soccer performance variables (running performance and game performance indicators) and the final game outcome (observed as loss, draw, win).

Data on the running performance of the players were collected by GPS technology (Catapult S5 and X4 devices, Melbourne, Australia) with a sampling frequency of 10 Hz. Such device was already investigated for metrics, and was found to be appropriately reliable and valid in sport settings (i.e., less than 1% measurement error, and 80% of common variance with running speed measured by timing gates) [31,32].

The variables included the following: total distance covered (m); distance in five speed categories (walking (<7.1 km/h), jogging (7.2–14.3 km/h), running (14.4–19.7 km/h), high-speed running (19.8–25.1 km/h), and maximal sprinting (>25.2 km/h)); number (frequency) of total accelerations (>0.5 m/s^2^); number of high-intensity accelerations (>3 m/s^2^); number of total decelerations (less than –0.5 m/s^2^) and number of high-intensity decelerations (less than –3 m/s^2^).

The game performance indicators for each player were determined by the position-specific InStat index (InStat, Moscow, Russia). The InStat index is calculated on the basis of a unique set of key parameters for each playing position (12–14 performance parameters, depending on the position during the game), with a higher numerical value indicating better performance. The exact calculations are trademarked and known only to the manufacturer of the platform. In most general terms, an automatic algorithm considers the player’s contribution to the team’s success, the significance of their actions, opponent’s level and the level of the competition they play in (i.e., the same performance done in European Champions League and some national-level first division will not be rated with same values). The rating is created automatically, and each parameter has a factor which changes depending on the number of actions and events in the match. The weight of the action factors differs depending on the player’s position. For example, grave mistakes done by CD and their frequency affect InStat index to a greater extent than those done by FWD. The key factors included in the calculation of the InStat index are position specific and include tackling, aerial duels, set pieces in defense, interceptions (for CD); number of crosses, number of passes to the penalty area, pressing (for FB); playmaking, number of key passes, finishing (for CM); pressing, dribbling, finishing, counterattacking (for WM); shooting, finishing, pressing, dribbling (for FWD). In order to calculate the InStat Index, the player has to spend a certain amount of time on the field and perform a minimum number of actions, but in this study this issue was solved simply by including only those players who played the whole game (as explained in Section 2.1).

### 2.3. Statistics

The normality of the distributions was checked by the Kolmogorov–Smirnov test, and the data are presented as the means ± standard deviations. The homoscedasticity of all variables was confirmed by Levene’s test. The statistical analyses were performed throughout several phases. 

In the first phase the data obtained by InStat index were associated with final game outcome by one-way analysis of variance (ANOVA). For this procedure the game outcomes (loss, draw, win) were considered as the grouping (independent) variable, and differences were established for total sample of players, and separately for each playing position. This allowed identification of the validity of the InStat index as an indicator of the final game achievement for the total sample, and for the five observed playing positions. 

The second phase of data analysis comprised calculation of differences among playing positions in running performance and InStat index. This was done by ANOVA with a consecutive Scheffe post hoc test. Throughout these analyses the information of running performance specifics for each playing position were obtained. Also, the analysis of differences in InStat allowed identification of the applicability of the InStat index for the analysis of game achievement for each playing position. To evaluate the effect sizes (ES), partial eta-squared values (η^2^) were presented (small ES: >0.02; medium ES: >0.13; large ES: >0.26) [33]. 

In the third phase, the associations between running performance (obtained by GPS) and game performance indicators (evaluated by InStat) were identified by calculating Pearson’s product moment correlation coefficients. 

For all analyses, Statistica 13.0 (TIBCO Software Inc., Greenwood Village, CO, USA) was used, and a *p* < 0.05 was applied.

## 3. Results

The ANOVA indicated significant (*p* < 0.05) association between the InStat index and match outcome for the total sample (*n* = 101, F-test: 23.69, η^2^ = 0.30 (large E)), CD (*n* = 26, F-test: 3.89, η^2^ = 0.24 (medium ES)), FB (*n* = 24, F-test: 4.98, η^2^ = 0.31 (large ES)) and CM (*n* = 33, F-test: 15.71, η^2^ = 0.50 (large ES)). The InStat index was not significantly associated with game outcomes for WM (*n* = 10, F-test: 0.98, η^2^ = 0.21 (medium ES)), and FW (*n* = 8, F-test: 2.61, η^2^ = 0.52 (large ES)) (Figure 2).

The descriptive parameters for running performances and InStat index in total sample, and for each playing positions are presented in Table 2. Significant ANOVA differences were found among playing positions (*p* < 0.05) in all running performances, with large ES for differences in: (i) total distance covered (η^2^ = 0.59); (ii) distance covered while jogging (η^2^ = 0.41); (iii) running (η^2^ = 0.62); (iv) high-speed running (η^2^ = 0.53); (v) sprinting (η^2^ = 0.39); (vi) number of performed accelerations (η^2^ = 0.27); (vii) number of decelerations (η^2^ = 0.45); (viii) number of high-intensity accelerations (η^2^ > 0.30); and (ix) number of high-intensity decelerations (η^2^ = 0.41). Small ES was found for differences in distance covered while walking (η^2^ = 0.11) (Table 3). 

Specifically, CM covered the longest total distance (significant post-hoc differences when compared to all other playing positions), the longest distance in jogging (significant post-hoc differences when compared to all other playing positions), and the longest distance while running (significantly different from CD and FB). WM covered the longest distance in high-speed running, and in sprinting (significant post-hoc differences to CD, CM, and FW). CD carried out the highest number of accelerations and highest number of decelerations (significantly different from FW). Finally, FW carried out the highest number of high-intensity accelerations (significant post-hoc differences when compared to CD, FB, and WM) and high-intensity decelerations (significantly different to WM) (Table 3). 

The total running distance and high-intensity accelerations were correlated with the InStat index for CD (r = 0.42 and r = 0.49, respectively). Furthermore, the number of decelerations was significantly correlated with the InStat in FB (r = –0.43), while distance covered during sprinting was correlated with InStat index in FW (r = 0.80). In general, the running performances of players in central and wide midfield positions were not significantly associated with the InStat index (Table 4).

## 4. Discussion

With regard to study aims there are two most important findings. First, the total distance covered and the intensity of running varied according to the different playing positions. Second, running performance parameters (e.g., the number of accelerations or decelerations and the distance covered in different speed zones) affect successful performance in soccer for some playing positions. Prior to discussion of these findings, an overview of the analyses done in order to evaluate the applicability and validity of InStat index as a measure of final match outcome will be provided.

Studies have already investigated the association between different variables explaining situational efficacy (i.e., game performance indicators) and match outcomes. For example, when losing the game, teams had more ball possession [30,34,35] and performed more crosses and dribbles [27]. Additionally, when winning, the teams performed more interceptions, clearances and aerial challenges, fewer passes and dribbles [27], and less high-intensity activities [18,34]. However, previous studies regularly investigated the performance indicators of the whole team, while there has been limited research investigating the position-specific performances in relation to game outcome, even though technical indicators have been considered good predictors of soccer match success [36]. Also, the quality of technical skills in real-game performance, which is actually obtained throughout the InStat index and other similar platforms, has been included as a main component in soccer talent identification and development systems [37,38]. 

InStat index in soccer is based on wide range of team- and individual-statistics, which are linked to the supporting video episodes. At the final stage, the calculated index should be related to final game outcome, and consequently should be a valid measure of final team achievement (i.e., game outcome). Results of this study indicated significant differences among game outcomes (loss, draw, or win) in InStat index for the total sample and specifically for CD, FB and CM. Although the statistical significance of the F-test did not reach statistical significance for WM and FW, this may be attributed to small number of players in these groups (WM: 10 players, FW: 8 players) and consequent small number of degrees of freedom [39]. Therefore, it might be said that the results presented here confirmed the validity of InStat in evaluation of final game achievement in Croatian professional soccer. It is also important to note that InStat index is specifically calculated for different positions on the basis of position-specific parameters (please see Section 2 for more details). Therefore, the lack of differences among playing positions in InStat (please see Table 2 and Table 3 for more details) indicates that this index might be observed as an applicable measure of position-specific game performance in soccer. 

### 4.1. Running Performances and Differences Among Playing Positions

Considering the different tactical roles of different playing positions in soccer games, recent studies confirm that the distance covered during the match appears to be related to playing position [11,14,16,20]. Results of this study evidenced significant differences in running performance among playing positions, and such results are generally in agreement with previous studies that investigated these issues in the English Premier League, the Spanish first division, the Italian Serie A, the French League 1, and the Brazilian first division [13,14,16,20,40]. Specifically, analysis of the Brazilian first division evidenced that the total distance covered by FB, CM and WM was greater than that covered by CD and FW [13]. Supporting this, the lowest total distance was found for CD (9313 m on average). At the same time, CM covered significantly more distance than players in all other positions (11,155 m, on average), which is known to be related to specific playing duties (i.e., CM are responsible for the connection between defense and attack, and such tactical roles require them to achieve greater distances) [14,16].

Previous studies performed indicated 10.7 km as the average total distance covered in Spanish and English top divisions [10,40,41]. Meanwhile players observed herein covered total distance of 10.3 km in average. Therefore, it seems that the total distance covered is not the factor that distinguishes Croatian players from those playing in elite European divisions. On the other hand, there is an evident difference in the intensity of running. More precisely, top-level European soccer players cover 10% of the total distance at a high intensity, which includes high-speed running and maximal sprinting [17,42]. Meanwhile, here presented results indicated that Croatian players perform 6.4% of the total distance covered at a high-intensity running pace. 

It is generally accepted that low-intensity activities, such as walking and jogging, are not crucial in elite soccer performance [43]. However, knowledge of these indicators is important to properly understand the position-specific demands. Thus, considering the percentage of the total distance, the most time spent walking and jogging is observed in CD (an average of 85.2% (7935 m) of their total distance covered (9313 m)). On the other hand, the least time spent walking and jogging is observed in WF (76.3%), followed by CM (79.4%), FB (79.8%) and FW (81.8%). Collectively, these findings support previous considerations that Croatian first division players generally play at a lower game pace when compared to elite European national division players, who spend a much lower percentage of time in low-intensity activities (from 74.9% to 79.6% of total distance) [10].

The distance covered while jogging among CM is significantly higher than for any other position. As mentioned before, CM had the greatest total distance, which is directly influenced by the distance covered while jogging. Furthermore, CM have the greatest distances in the “running zones” (e.g., 14.4–19.7 km/h). Therefore, results support the findings from previous studies in which authors reported similar figures and concluded that the physical performance of CM is characterized by covering a high overall distance, especially at moderate to high speeds such as jogging and running [10].

High-intensity activities are usually defined as all activities with running speeds of 19.8 km/h and above, and the distance covered at high intensities has been traditionally identified as a key performance indicator of physical match performance [44] and one of the crucial elements of success in soccer [43]. The results showed that the greatest amount of high-intensity running (high-speed running + sprint running) is covered by WM, while the CD have the lowest values for these indicators. This is consistent with previous investigations in which authors reported similar results for the English Premier League and the Spanish first division [10,14,20,40].

It is known that outside players (e.g., WM and FB) perform significantly more sprints than players in central playing positions [14]. Supporting this, our results showed that the greatest sprint distance was covered by WM and FB. However, despite similar differences among playing positions between our study and previous studies, values of high-intensity running in Croatian players were evidently lower than those from the best European national competitions [10,40]. More specifically, the mean high-intensity distance covered among all playing positions in the English Premier League was 936 m, in the Spanish first division an average of 821 m was reported, while the average value for high-intensity running in Croatian players was 652 m.

The highest number of accelerations and decelerations was found for CD and the lowest for FW, which is consistent with some similar studies on friendly matches in the Spanish first division [10]. Specifically, one of the most important tactical roles of FW is to keep the ball in possession in the central position, so it is expected that FWs do not cover a large distance. On the other hand, CD must be constantly prepared for defensive reactions. While trying to find appropriate positioning, they frequently change running directions, but also the type of running (i.e., frontal running to make a defensive line to catch opposing players in offsides and lateral shuffles to obtain better positions versus FW). This certainly results in a high number of accelerations and decelerations for CD. However, the kind of accelerometer unit and the way that the data are mathematically treated could have a significant effect on the calculation of accelerations and decelerations, which actually limits the comparability between different studies [10]. Specifically, while the capacity to accelerate and decelerate plays a critical role in elite soccer, as it represents high energy demanding activities, the determination of accelerations might still have unresolved methodological issues [10].

### 4.2. Associations between Game Performance Indicators and Running Performances— Playing Position Approach

The results suggest that CD covered the shortest distance while running out of all playing positions, and this is in agreement with previous reports where authors found that CD exert the fewest high-intensity efforts compared to all other playing positions [10,14,20,40]. This is understandable knowing that their technical roles (i.e., aerial duels, tackles, positioning, and interceptions of the balls passed to the attackers) are generally more focused on the reactions or accelerations and then on high-speed running. As a result, most of their high-intensity efforts are performed in the zone of running (14.4–19.7 km/h) simply because they do not have many opportunities to develop running speeds above the high-intensity zone threshold (>19.8 km/h). However, because of the positive correlation between the InStat index and the distance covered while running (14.4–19.7 km/h) for CD, running should be considered an important determinant that affects success for this position. Furthermore, a positive correlation between the numbers of high-intensity accelerations and the InStat index among CD shows that a greater number of high-intensity accelerations directly affects real game performance for this playing position. Specifically, stepping out to the duels and putting pressure on opponent players are two of the most important tactical roles of CD. If performed rapidly and aggressively (in other words, with a high acceleration), the chances of winning a duel increase, which consequently has positive repercussions on final match achievement as well.

The total number of decelerations was inversely associated with the InStat index among FB, meaning that a higher number of decelerations negatively affected real game efficacy for that playing position. Although FB are basically defensive players and their starting tactical line-up is in the first third of the pitch, the main technical requirements for FB are the number of entries to the third part of the pitch (i.e., pressing) and the number of crosses [38,45]. These duties are actually performed on the opponent’s half. Therefore, some of the most important tactical roles of the FB are actually in attacking. To create more of these activities, FB frequently have to move away from the starting tactical line-up, which actually enables them to make crosses and press. Consequently, if FB have a higher number of stoppings (i.e., decelerations), it probably negatively affects their ability to participate in attacking actions and to perform crosses and entries to the third part of the pitch. Collectively, it seems that soccer success of FB is more affected by their attacking activities, regardless of the fact that they are defensive players.

Previously, it was highlighted that FW had evidently shorter sprinting distances than players in the same position during games from other European competitions (please see previous discussion for details). However, results indicated a strong correlation between the InStat index and the sprint distance for this playing position, which led to conclusion that the sprint distance covered during the game was a highly important determinant of overall game performance for FW. Indeed, FW are positioned close to the opponent’s goal, and almost every sprint presents the opportunity to perform attacking actions. In addition, FW have the lowest number of tackles, interceptions and clearances compared with other playing positions [38], which suggests that most of their activities are focused on attacking. With the higher number of attacking actions, there is a growing chance to enter the penalty area, shoot, and score. As a result, the number of attacking situations increases the likelihood for positive game outcomes [28,29,30].

The main role of CM is to organize the offense by proper ball control and passes, rather than by invasion into the opponent’s area [38]. Considering the lack of significant association between running parameter and the InStat index for this playing position, it seems that CM soccer success is more influenced by some variables, other than those obtained by GPS, such as ball possession, number of key passes, dribbles, and shots. Also, the running indicators obtained by GPS measurements were not correlated to InStat variables in WM, which may be observed as surprising since WM experience the greatest physical requirements during the game, both in terms of total distance covered and high-intensity running [10]. The possible explanation may be the previously discussed finding of the small amount of overall distance covered in the studied Croatian players, which actually resulted in truncated variance and consequently statistically/mathematically decreased the possibility of achieving significant correlations.

### 4.3. Limitations and Strengths

The main limitation comes from the fact that this study observed only one team which was observed during one half season. Therefore, some specific covariates (limited number of observed players, strength of the opponent, specific tactical requirements) may influence reported results. Next, in this study no data were collected about psycho-physiological responses of the players (e.g., heart rate and RPE), which are known to be important determinants of overall performance. Further, this study actually studied relatively simple “game-related outcomes” (i.e., running performances obtained by GPS and game performance indicators obtained by InStat index), while sport-performances, especially those in team sports, are far more complex (i.e., include interaction, cooperation, and opposition) [46]. Also, in this study relatively simple methodology was applied, while complex systems like sport games may ask for more detailed experimental approaches and the use of mixed methods as an observational methodology [47]. 

On the other hand, this study has several strengths. First, this is one of the first studies which simultaneously evaluated two sets of performance variables (i.e., running performances and game performance indicators) and probably the first one where associations between these two groups of performances were analyzed. Also, the data were collected during official games, among professional players, and at the highest national competitive level. Therefore, results are generalizable to similar samples of participants and levels of competition. Furthermore, the position-specific approach in identification of the relationships between running performances and game performance indicators is important strength of the investigation. Therefore, despite the evident limitations, the authors believe that this study may contribute to the knowledge on this field and initiate further research. 

## 5. Conclusions

The total distance covered during the match did not distinguish Croatian first division players from players who compete in elite European divisions (i.e., Spain and Germany). However, the players studied here achieved total distances at lower running speeds than their peers involved in top-level European competitions, which clearly indicates the lower game pace in Croatian soccer competition. These findings can be useful for determining the physical requirements and profiles of the players in the Croatian first division, especially with regard to international competitions (i.e., the European League, Champions League).

This study confirmed the association between the running performance of players involved in certain playing positions and overall game performance. Specifically, it seems that CD distance in the running zone and number of high-intensity accelerations, FB number of decelerations, and FW sprinting distance are crucial physical requirements of team success.

Training prescriptions in soccer should be based on established requirements specific to the playing positions, thereby ensuring that players are more able to fulfill their game duties and tactical responsibilities over the soccer match. In further studies it would be important to identify possible associations that might exist between different parameters of players’ conditioning status and indicators of real game performance. 

## Figures and Tables

**Figure 1 ijerph-16-04032-f001:**
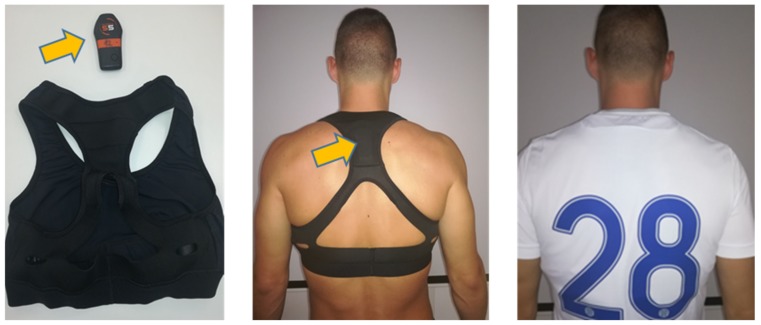
Global positioning device (GPS) used for the measurement of running performances in soccer.

**Figure 2 ijerph-16-04032-f002:**
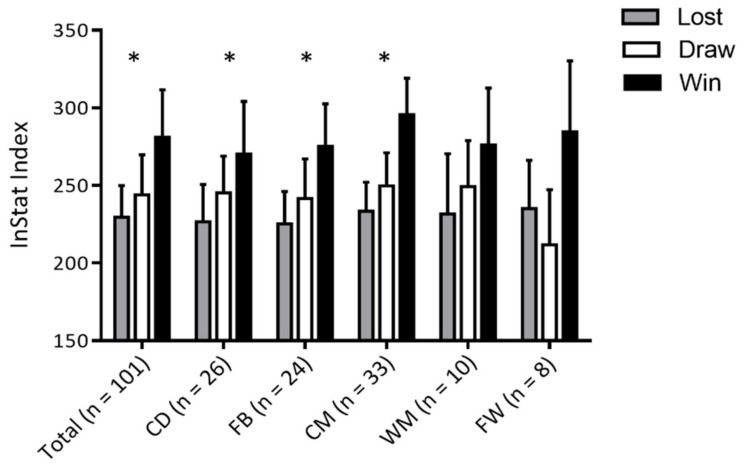
InStat index in relation to the outcome of the match for total sample (Total) and different playing positions (CD, central defenders; FB, full-backs; CM, central midfielders; WM, wide midfielders; FW, forwards); * indicates statistically significant differences at *p* < 0.05 derived by analysis of variance

**Table 1 ijerph-16-04032-t001:** Sociodemographic and anthropometric characteristics of the studied players with differences among playing positions (F-test).

	Age (years)	Body height (cm)	Body mass (kg)
	Mean ± SD	Mean ± SD	Mean ± SD
Total sample (*n* = 101)	23.85 ± 2.88	183.05 ± 6.88	78.69 ± 7.17
Central Defenders (*n* = 26)	23.25 ± 2.21	192.25 ± 5.61	87.27 ± 7.38
Full-Backs (*n* = 24)	23.2 ± 3.56	176.6 ± 3.36	73.4 ± 4.34
Central Midfielders (*n* = 33)	22.66 ± 2.73	175 ± 6.08	76.51 ± 5.02
Wide Midfielders (*n* = 10)	26.0 ± 1.0	183 ± 3.46	76.2 ± 4.17
Forwards (*n* = 8)	27.0 ± 2.82	181.5 ± 0.7	85 ± 4.1
F-test (*p*)	1.51 (0.24)	5.92 (0.01)	5.04 (0.01)

**Table 2 ijerph-16-04032-t002:** Descriptive statistics for running performances and game performance indicator (InStat).

Variables	Total	Central Defenders	Full-Backs	Central Midfielders	Wide Midfielders	Forwards
	Mean ± SD	Mean ± SD	Mean ± SD	Mean ± SD	Mean ± SD	Mean ± SD
Total distance (m)	10,298.4 ± 928.68	9313.5 ± 599.4	10,368 ± 612	11,155.1 ± 635.3	10,264.8 ± 275.2	9796.7 ± 703.7
Walking (m)	4220.57 ± 362.33	4076.6 ± 378.3	4297.9 ± 338.5	4258.5 ± 340.7	4074.8 ± 194.3	4482.1 ± 442.2
Jogging (m)	4092.94 ± 569.73	3859 ± 380.2	3975.4 ± 372.8	4599.7 ± 471.4	3761.2 ± 324.1	3530 ± 729.9
Running (m)	1363.27 ± 339.68	999.2 ± 197.7	1320.7 ± 236.1	1674.9 ± 226.1	1526.5 ± 117.4	1184.4 ± 207.9
High-speed running (m)	461.83 ± 160.15	288.2 ± 63.8	533.9 ± 134.1	492.7 ± 139.9	640.7 ± 105.4	458.7 ± 94.7
Sprinting (m)	155.89 ± 97.13	87.7 ± 59.9	236.6 ± 97.2	123.7 ± 69.5	260.6 ± 68.8	137.1 ± 46.9
Accelerations (count)	716.19 ± 73.15	743.5 ± 56.2	710 ± 66.2	733.4 ± 72.4	688 ± 34.2	610.1 ± 83.7
Decelerations (count)	674.44 ± 69.29	714.1 ± 51.5	672.4 ± 56	681.9 ± 55.8	661.8 ± 36.7	536.6 ± 69
High-intensity accelerations (count)	3.16 ± 2.67	2.5 ± 1.8	3.1 ± 1.7	1.9 ± 2.2	7 ± 2.6	6 ± 2.9
High-intensity decelerations (count)	11.39 ± 6.27	6.1 ± 2.8	13.1 ± 4.9	11.5 ± 5.9	20.8 ± 5.5	11 ± 3.1
InStat (index)	284.5 ± 31.04	247.4 ± 29.2	243 ± 28.7	254.1 ± 29.3	251.1 ± 32.1	242 ± 49.5

**Table 3 ijerph-16-04032-t003:** Differences among playing positions for running performances and game performance indicator (InStat) determined by analysis of variance (ANOVA), with Scheffe post-hoc test differences.

**Variables**	**ANOVA**	**Effect Size**	**Post hoc**				
	**F (p)**	η^2^	**Central Defenders**	**Full-Backs**	**Central Midfielders**	**Wide Midfielders**	**Forwards**
Total distance (m)	35.02 (0.01)	0.59	^FB, CM, WM^	^CD, CM^	^CD, FB, WM, FW^	^CD, CM^	^CM^
Walking (m)	3.18 (0.02)	0.11	-	-	-	-	-
Jogging (m)	16.71 (0.01)	0.41	^CM^	^CM^	^CD, FB, WM, FW^	^CM^	^CM^
Running (m)	39.30 (0.01)	0.62	^FB, CM, WM^	^CD, CM^	^CD, FB^	^CD, FW^	^CM^
High-speed running (m)	29.30 (0.01)	0.53	^FB, CM, WM, FW^	^CD^	^CD, WM^	^CD, CM, FW^	^CD, WM^
Sprinting (m)	15.72 (0.01)	0.39	^FB, WM^	^CD, CM, FW^	^FB, WM^	^CD, CM, FW^	^FB, WM^
Accelerations (count)	9.06 (0.01)	0.27	^FW^	^FW^	^FW^		^CD, CM, FW^
Decelerations (count)	20.11 (0.01)	0.45	^FW^	^FW^	^FW^	^FW^	^CD, FB, CM, WM^
High-intensity accelerations (count)	8.53 (0.01)	0.30	^WM, FW^	^WM, FW^	^WM, FW^	^CD, FB, CM^	^CD, FB, WM^
High-intensity decelerations (count)	16.70 (0.01)	0.41	^FB, CM, WM^	^CD, WM^	^CD, WM^	^CD, FB, CM, FW^	^WM^
InStat (index)	0.64 (0.62)	0.03	-	-	-	-	-

Superscripted letters indicate significant post-hoc differences when compared to specific playing position (CD, central defenders; FB, full-backs; CM, central midfielders; WM – wide midfielders; FW, forwards).

**Table 4 ijerph-16-04032-t004:** Pearson’s product moment correlations between running performances and game performance indicator (InStat) for different playing positions.

**Variables**	**Total (*n* = 101)**	**Central Defenders (*n* = 26)**	**Full-Backs (*n* = 24)**	**Central Midfielders (*n* = 33)**	**Wide Midfielders (*n* = 10)**	**Forwards (*n* = 8)**
Total distance	0.08	0.18	–0.04	–0.02	–0.17	0.01
Walking	−0.02	0.09	0.01	–0.12	0.07	0.04
Jogging	0.05	0.02	–0.10	0.05	–0.41	–0.05
Running	0.16	0.42 *	0.02	0.12	0.01	–0.13
High-speed running	0.02	–0.04	–0.06	–0.10	0.54	0.17
Sprinting	0.01	–0.24	0.17	–0.04	0.22	0.80 *
Accelerations	–0.01	0.12	–0.39	0.07	–0.24	–0.02
Decelerations	–0.09	0.07	–0.43 *	–0.05	–0.26	–0.33
High-intensity accelerations	0.18	0.49 *	0.20	0.29	–0.08	0.26
High-intensity decelerations	0.05	0.29	–0.04	–0.01	0.44	–0.18

* denotes statistical significance of *p* < 0.05.

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
