# Peer review of "Analysis of the Association between Running Performance and Game Performance Indicators in Professional Soccer Players"

_ijerph, 2019, doi:10.3390/ijerph16204032_

Round 1
Reviewer 1 Report
Although this is a well documented manuscript, it is very difficult to follow. The sentences are to long and the English language needs serious improvements.
Introduction
Lines 38-39, 40 - Please rephrase. The sentences are ambiguous. The words achievement and performance are used twice in the same sentence.
Line 56 - instead of ”allows data to be collected about players`running performance”, you should consider ”allows collecting data about players`running performance”.
Line 96 - please rephrase ”we were of the opinion...!
Material and method
You should add age, height, weight for each group. Are there any differences based on playing position?
Please provide some data about InStat index (how is it calculated, how are the parameters recorded, accuracy, etc).
Line 134 - please quote the p-value as 0.05
The ethical statement is missing. Please follow the journal`s requirement regarding Research Involving Human Subjects.
Results
The data presented in lines 144-156 are very difficult to follow.
Discussion
Line 180 - The authors reported that InStat index was a valid measure of soccer success. There are no data in the manuscript sustaining this fact. Please explain.
Lines - 196-198 - Please rephrase. The sentence is too long and needs serious English improvement.
Line 206 - instead of ”...is by CD” , ”...is observed in CD”, for example. The same in line 208.
Author Response
Dear Sir/Madam
First of all, let us express our gratitude for your constructive and elaborated review. We have tried to follow it strictly. Please find below how we responded to your comments.
Staying at your disposal
Authors
Although this is a well documented manuscript, it is very difficult to follow. The sentences are to long and the English language needs serious improvements.
RESPONSE: Thank you for recognizing the potential of our manuscript. We have tried to follow all your comments and amended the manuscript accordingly. Also, special attention is paid on language quality and grammar. Please find bellow how we corrected the manuscript following your suggestions
Introduction
Lines 38-39, 40 - Please rephrase. The sentences are ambiguous. The words achievement and performance are used twice in the same sentence.
RESPONSE: Text is amended and now reads: „ Soccer is a highly complex team sport with changing dynamics and multistructural movements played by two teams. Each team consists of 10 outfield players and a goalkeeper and the final game achievement depends directly on the performance of all 11 players [1,2]., Therefore, performance analysis is crucial in the evaluation of players’ achievement [3]. (Please see begining of the introduction)
Line 56 - instead of ”allows data to be collected about players`running performance”, you should consider ”allows collecting data about players`running performance”.
RESPONSE: Amended accordingly. Text reads: „ Specifically, GPS allows collecting data about players’ running performance, such as the total distance covered, the distance covered at different intensities (e.g., speeds), the number of accelerations and decelerations, etc.”
Line 96 - please rephrase ” we were of the opinion...
RESPONSE: Rephrased, thank you.
Material and method
You should add age, height, weight for each group. Are there any differences based on playing position?
RESPONSE: Thank you for this suggestion. Anthropometric and sociodemographic data of players according to playing position are now presented in (new) Table 1.
Please provide some data about InStat index (how is it calculated, how are the parameters recorded, accuracy, etc).
RESPONSE: The text explaining the calculation of the InStat index is included and reads: “The game performance indicators for each player were determined by the position-specific InStat index (InStat, Moscow, Russia). The InStat index is calculated on the basis of a unique set of key parameters for each playing position (12 to 14 performance parameters, depending on the position during the game), with a higher numerical value indicating better performance. The exact calculations are trademarked and known only to the manufacturer of the platform. In most general terms, an automatic algorithm considers the player’s contribution to the team’s success, the significance of their actions, opponent’s level and the level of the competition they play in (i.e. same performance done in European Champions League and some national-level first division will not be rated with same values. The rating is created automatically, and each parameter has a factor which changes depending on the number of actions and events in the match. The weight of the action factors differs depending on the player’s position. For example, grave mistakes done by CD and their frequency affect InStat Index to a greater extent than those done by FWD. The key factors included in the calculation of the Instat Index are position specific and include tackling, aerial duels, set pieces in defense, interceptions (for CD); number of crosses, number of passes to the penalty area, pressing (for FB); playmaking, number of key passes, finishing (for CM); pressing, dribbling, finishing, counterattacking (for WM); shooting, finishing, pressing, dribbling (for FWD). In order to calculate the InStat Index, the player has to spend a certain amount of time on the field and perform a minimum number of actions.”
Line 134 - please quote the p-value as 0.05
RESPONSE: Amended accordingly.
The ethical statement is missing. Please follow the journal`s requirement regarding Research Involving Human Subjects.
RESPONSE: Thank you for noticing it. Text explaining the ethical statement is added and reads: „The investigation was approved by Ethical Board of the University of Split, Faculty of Kinesiology, Split, Croatia (approval number: 2181-205-02-05-19-0020).”
Results
The data presented in lines 144-156 are very difficult to follow.
RESPONSE: We must agree that the text where results were explained was difficult to follow. Therefore, this part of the text is systematically rewritten and we believe that it is more fluent. It reads: The descriptive parameters for running performances and InStat index in total sample, and for each playing positions are presented in Table 2. Significant ANOVA differences among playing positions (p < 0.05) in all running performances, with large ES for differences in: (i) total distance covered (η2 = 0.59), (ii) distance covered while jogging (η2 = 0.41), (iii) running (η2 = 0.62), (iv) high speed running (η2 = 0.53), (v) sprinting (η2 = 0.39), (vi) number of performed accelerations (η2 = 0.27), (vii) number of decelerations (η2 = 0.45), (viii) number of high intensity accelerations (η2 > 0.30), and (viii) number of high intensity decelerations (η2 = 0.41). Small ES was found for differences in distance covered while walking (η2 = 0.11) (Table 3). Specifically, CM covered longest total distance (significant post-hoc differences when compared to all other playing positions), longest distance in jogging (significant post-hoc differences when compared to all other playing positions), and longest distance while running (significantly different from CD and FB). WM covered longest distance in high speed running, and in sprinting (significant post-hoc differences to CD, CM and FW). CD carried out the highest number of accelerations, and highest number of decelerations (significantly different from FW). Finally, FW carried out the highest number of high intensity accelerations (significant post-hoc differences when compared to CD, FB, and WM), and high intensity decelerations (significantly different to WM) (Table 3).”
Also, we hope that understanding was improved by dividing table of statistics and ANOVA calculations in two (now Table 2 and 3). Thank you.
Discussion
Line 180 - The authors reported that InStat index was a valid measure of soccer success. There are no data in the manuscript sustaining this fact. Please explain.
RESPONSE: We believe that idea of InStat validation is now more clearly described in the Statistics subsection and later in Discussion. Text in Statistics subsection reads: „In the first phase the data obtained by InStat index were associated with final game outcome by one-way analysis of variance (ANOVA). For this procedure the game outcomes (lost – draw – win) was considered as grouping (independent) variable, and differences were established for total sample of players, and separately for each playing positions. It allowed identification of predictive validity of InStat index as an indicator of the final game achievement.” (Please see 3rd paragraph of the Statistics subsection. Thank you)
Lines - 196-198 - Please rephrase. The sentence is too long and needs serious English improvement.
RESPONSE: Thank you. This part of the text is rewritten and now reads:“ Previous studies indicated 10.7 km as the average total distance covered in Spanish and English top divisions [1-3]. Meanwhile players observed herein covered total distance of etc.“ (Please see paragraph of the subsection 4.1)
Line 206 - instead of ”...is by CD” , ”...is observed in CD”, for example. The same in line 208.
RESPONSE: Amended accordingly. Thank you.
Thank you once again. Authors
Reviewer 2 Report
Title: 2-4
The title should respond to the main objective. Include another title please.
Introduction: 95-98
In scientific studies should avoid the use of personal forms. It must be corrected all over the paper.
‘Additionally, we compared running performance and standard soccer performance variables among playing positions. We were of the opinion that a study of this type would allow a better understanding of the relationships that exist between running performance and game performance indicators and that therefore would improve the applicability of both sets of variables in soccer training and competition.’
Procedures: 110-121
- Procedure is incomplete. Please, readers needs more information (step by step).
- What ‘12 to 14 performance parameters have been used’? They should be included.
- Please, provide any evidence of reliability to GPS technology (Catapult S5 112 and X4 devices, Melbourne, Australia) and InStat index (InStat, Moscow, Russia).
Statistics: 129-132
-I consider this paragraph is highly questionable. The process of validation needs a greater specificity.
‘The association between the InStat index and final game outcome was determined by ANOVA with the final game outcome as a grouping variable (lost – draw – win), which allowed us to identify the VALIDITY OF THE INSTAT INDEX.’
- I guess you are aware of the risks of using log-transform data. I suggest, the analyzes should be repeated to consider whether results became different. I wish I could know more about the final results. I would suggest I can review the new analyzes before I can fully accept this work.
- I miss the effect sizes (Cohen-d) in the data analysis section. Please, must be done.
https://www.ncbi.nlm.nih.gov/pmc/articles/PMC4120293/
Discussion: 180
-I remind you the comment about Validity.
‘Second, the InStat index was a valid measure of soccer success’.
Limitations and strengths: 312
-I am sure the authors will find many more applications that make the article more necessary.
-I consider that this work has two main limitations more. In science a theoretical framework is necessary. I suggest that two limitations be included, one, to use the praxiology of Pierre Parlebas, as an example of a theoretical framework. I have made a recent selection you should used (Parlebas, 2018; Pic et al 2019)
Parlebas, P. (2018). Une pédagogie des compétences motrices. Acción Motriz, 20(1), 89- 96
Pic, M., Lavega-Burgués, P., & March-Llanes, J (2019). Motor behaviour through traditional games, Educational Studies, 45:6, 742-755, DOI: 10.1080/03055698.2018.1516630
On the other hand, the use of an mixed methods as a observational methodology (Anguera et al 2018; Casal et al; Pic, 2018) due to the possibility to contextualize the results against completely impersonal data (studies). Please, include next references to complete the work.
Anguera, M. T., Blanco-Villaseñor, A., Losada, J. L., Sánchez-Algarra, P., and Onwuegbuzie, A. J. (2018b). Revisiting the difference between mixed methods and multimethods: is it all in the name? Qual. Quant. 52, 2757–2770. doi: 10.1007/s11135-018-0700-2
Pic, M. (2018). Performance and home advantage in handball. J Hum Kinet, 63(1), 61-71.
Casal CA, Anguera MT, Maneiro R and Losada JL (2019) Possession in Football: More Than a Quantitative Aspect – A Mixed Method Study. Front. Psychol. 10:501. doi: 10.3389/fpsyg.2019.00501
Author Response
Dear Sir/Madam
Thank you for recognizing the potential of our work. We have found all your comments as highly valuable and amended the manuscript accordingly. Please see below how we responded and where to find specific changes.
Staying at your disposal
Authors
Title: 2-4
The title should respond to the main objective. Include another title please.
RESPONSE: The title is changed, and now reads: “Analysis of the association between running performance and game performance indicators in professional soccer players“ Thank you
Introduction: 95-98
In scientific studies should avoid the use of personal forms. It must be corrected all over the paper.
‘Additionally, we compared running performance and standard soccer performance variables among playing positions. We were of the opinion that a study of this type would allow a better understanding of the relationships that exist between running performance and game performance indicators and that therefore would improve the applicability of both sets of variables in soccer training and competition.’
RESPONSE: The personal writing is avoided throughout the manuscript. Therefore, all sentences are transformed in „pasive voice“. Thank you.
Procedures: 110-121
- Procedure is incomplete. Please, readers needs more information (step by step).
RESPONSE: We believe that Procedures are now more precisely explained. In the first part, we explained Catapult system, its reliability and validity. In the next part derivation of the InStat index is accurately presented (please see next response)
- What ‘12 to 14 performance parameters have been used’? They should be included.
RESPONSE: In this version of the manuscript special attention is paid on precise explanation and presentation of the InStat index. Text reads: “The game performance indicators for each player were determined by the position-specific InStat index (InStat, Moscow, Russia). The InStat index is calculated on the basis of a unique set of key parameters for each playing position (12 to 14 performance parameters, depending on the position during the game), with a higher numerical value indicating better performance. The exact calculations are trademarked and known only to the manufacturer of the platform. In most general terms, an automatic algorithm considers the player’s contribution to the team’s success, the significance of their actions, opponent’s level and the level of the competition they play in (i.e. same performance done in European Champions League and some national-level first division will not be rated with same values. The rating is created automatically, and each parameter has a factor which changes depending on the number of actions and events in the match. The weight of the action factors differs depending on the player’s position. For example, grave mistakes done by CD and their frequency affect InStat Index to a greater extent than those done by FWD. The key factors included in the calculation of the Instat Index are position specific and include tackling, aerial duels, set pieces in defense, interceptions (for CD); number of crosses, number of passes to the penalty area, pressing (for FB); playmaking, number of key passes, finishing (for CM); pressing, dribbling, finishing, counterattacking (for WM); shooting, finishing, pressing, dribbling (for FWD). In order to calculate the InStat Index, the player has to spend a certain amount of time on the field and perform a minimum number of actions, but in this study this issue was solved simply by including only those players who played the whole game (please see previous text about Participants and design).
- Please, provide any evidence of reliability to GPS technology (Catapult S5 112 and X4 devices, Melbourne, Australia) and InStat index (InStat, Moscow, Russia).
RESPONSE: Data are included and text now reads: Data on the running performance of the players were collected by GPS technology (Catapult S5 and X4 devices, Melbourne, Australia) with a sampling frequency of 10 Hz. Such device was already investigated for metrics, and was found to be appropriately reliable and valid in sport settings (i.e. less than 1% measurement error, and 80% of common variance with running speed measured by timing gates) [4].
Statistics: 129-132
-I consider this paragraph is highly questionable. The process of validation needs a greater specificity.
RESPONSE: Thank you for noticing it. The section on statistics is systematically rewritten and we hope that it now includes all necessary details. Text reads (please see underlined text for details on validity): „The normality of the distributions was checked by the Kolmogorov-Smirnov test, and the data are presented as the means ± standard deviations. The homoscedasticity of all variables was confirmed by Levene’s test. The further statistical analyses were performed throughout several phases.
In the first phase the data obtained by InStat index were associated with final game outcome by one-way analysis of variance (ANOVA). For this procedure the game outcome (lost – draw – win) was considered as grouping (independent) variable, and differences were established for total sample of players, and separately for each playing positions. It allowed identification of predictive validity of InStat index as an indicator of the final game achievement.
The second phase of data analyses comprised calculation of differences among playing positions in running performance and InStat index. This was done by ANOVA with a consecutive Scheffe post hoc test. Throughout these analyses the information of running performance specifics for each playing position were obtained. Also, the analysis of differences in InStat allowed identification of applicability of InStat index for the analysis of game achievement for each playing position.
In the third phase, the associations between running performance (obtained by GPS) and game performance indicators (evaluated by InStat) were identified by calculating Pearson’s product moment correlation coefficients.
For all analyses, Statistica 13.0 (TIBCO Software Inc., USA) was used, and a p<0.05 was applied.”
‘The association between the InStat index and final game outcome was determined by ANOVA with the final game outcome as a grouping variable (lost – draw – win), which allowed us to identify the VALIDITY OF THE INSTAT INDEX.’
RESPONSE: The validity of the InStat index is now discussed in the 2nd and 3rd paragraph of the Discussion. Text reads: Studies have already investigated the association between different variables explaining situational efficacy (i.e. game performance indicators) and match outcomes. For example, when losing the game, teams had more ball possession [30,34,35] and performed more crosses and dribbles [27]. Additionally, when winning, the teams performed more interceptions, clearances and aerial challenges, fewer passes and dribbles [27], and less high-intensity activities [18,34]. However, previous studies regularly investigated the performance indicators of the whole team, while there has been limited research investigating the position-specific performances in relation to game outcome, even though technical indicators have been considered good predictors of soccer match success [36]. Additionally, the quality of technical skills in real-game performance, which are actually obtained throughout the InStat index and other similar platforms, has been included as a main component in soccer talent identification and development systems [37,38]. InStat index in soccer is based on wide range of team- and individual-statistics, which are linked to the supporting video episodes. At the final stage, the calculated index should be related to final game outcome, and consequently should be a valid measure of final team achievement (i.e. game outcome). Results of this study indicated significant differences among game outcomes (lost – draw - win) in InStat index, for total sample, and specifically for CD, FB and CM. Although the statistical significance of the F-test did not reach statistical significance for WM and FW, this may be attributed to small number of players in these groups (WM: 10 players, FW: 8 players) and consequent small number of degrees of freedom [39]. Therefore, it might be said that here presented results confirmed the validity of InStat in evaluation of final game achievement in Croatian professional soccer. It is also important to note that InStat index is specifically calculated for different positions on the basis of position-specific parameters (please see Methods for more details). Therefore, the lack of differences among playing positions in InStat (please see Tables 2 and 3 for more details) indicates that this index might be observed as an applicable measure of position-specific game performance in soccer.
- I guess you are aware of the risks of using log-transform data. I suggest, the analyzes should be repeated to consider whether results became different. I wish I could know more about the final results. I would suggest I can review the new analyzes before I can fully accept this work.
RESPONSE: Following your suggestion in this version of the manuscript non-log transformed dana were analyzed and results are presented accordingly. Thank you.
- I miss the effect sizes (Cohen-d) in the data analysis section. Please, must be done.
RESPONSE: Thank you for this suggestion. Since we calculated ANOVA differences among several groups for numerous variables, we presented effect size differences on the basis partial eta squared, and results are presented in Results section
Discussion: 180
-I remind you the comment about Validity.
‘Second, the InStat index was a valid measure of soccer success’.
RESPONSE: Please see previous comment and response where we already specified changes we have done accordingly. Thank you.
Limitations and strengths: 312
-I am sure the authors will find many more applications that make the article more necessary.
-I consider that this work has two main limitations more. In science a theoretical framework is necessary. I suggest that two limitations be included, one, to use the praxiology of Pierre Parlebas, as an example of a theoretical framework. I have made a recent selection you should used (Parlebas, 2018; Pic et al 2019)
Parlebas, P. (2018). Une pédagogie des compétences motrices. Acción Motriz, 20(1), 89- 96
Pic, M., Lavega-Burgués, P., & March-Llanes, J (2019). Motor behaviour through traditional games, Educational Studies, 45:6, 742-755, DOI: 10.1080/03055698.2018.1516630
On the other hand, the use of an mixed methods as a observational methodology (Anguera et al 2018; Casal et al; Pic, 2018) due to the possibility to contextualize the results against completely impersonal data (studies). Please, include next references to complete the work.
Anguera, M. T., Blanco-Villaseñor, A., Losada, J. L., Sánchez-Algarra, P., and Onwuegbuzie, A. J. (2018b). Revisiting the difference between mixed methods and multimethods: is it all in the name? Qual. Quant. 52, 2757–2770. doi: 10.1007/s11135-018-0700-2
Pic, M. (2018). [5]. J Hum Kinet, 63(1), 61-71.
Casal CA, Anguera MT, Maneiro R and Losada JL (2019) Possession in Football: More Than a Quantitative Aspect – A Mixed Method Study. Front. Psychol. 10:501. doi: 10.3389/fpsyg.2019.00501
RESPONSE: Thank you for your suggestion. The section on Limitations and strengths is systematically rewritten and now include some of the citations you suggested. Text now reads: “The main limitation comes from the fact that study observed only one team which was observed during one half season. Therefore, some specific covariates (limited number of observed players, strength of the opponent, specific tactical requirements) may influence reported results. Next, in this study no data were collected about psycho-physiological responses of the players (i.e., heart rate, RPE), which are known to be important correlates of overall performance. Further, this study actually studied single “game-related outcomes” (e.g. running performance obtained by GPS, and game performance indicators obtained by InStat index), while sport-performances especially in team sports are more complex (i.e. include interaction, cooperation, opposition) [6]. Also, in this study relative simple methodology was applied, while complex systems like sport games may ask for a more detailed experimental approaches and the use of mixed methods as a observational methodology [5]. On the other hand, this study has several strengths. First, this is one of the first studies which simultaneously evaluated two sets of performance variables (e.g., running performances and game performance indicators) and probably the first one where associations between these two groups of game performances were analyzed. Also, the data was collected during official games and the highest national competitive level. Therefore, results are generalizable to similar samples of participants and levels of competition. Further, position-specific approach in identification of the relationships between running performances and game performance indicators is important strength of the investigation Therefore, despite the clear limitations, authors believe that study may contribute to the knowledge on a field, and initiate further research. “
Staying at your disposal
Authors
Round 2
Reviewer 1 Report
I appreciate the Author`s effort to improve the text.
My questions and concerns have been answered.
Reviewer 2 Report
Thank you very much to the authors
for the effort made